# Cultural adaptation and psychometric properties of the 8-item Patient Health Questionnaire (PHQ-8) to screen for depression in southwestern Madagascar

## Research Article

depression; psychometric properties; Vezo; Masikoro; Madagascar; climate health; planetary health

**Corresponding author:**
Hervet Randriamady, MS;
Email: hrandriamady@g.harvard.edu

Hervet J. Randriamady[1,2,3] 🔾, Manasi Sharma[4], Rocky E. Stroud II[4], Aroniaina M. Falinirina[5], Romario[5], Madeleine Rasoanirina[5], Nadège V. Volasoa[6], Frédéric Déclerque[5], Marc Y. Solofoarimanana[5], Jean C. Mahefa[5], Hanitra O. Randriatsara[7], Karestan C. Koenen[4,8] and Christopher D. Golden[2,3,9,10]

[1]Harvard Kenneth C. Griffin Graduate School of Arts and Sciences, Cambridge, MA, USA; [2]Department of Nutrition, Harvard TH Chan School of Public Health, Boston, MA, USA; [3]Madagascar Health and Environmental Research (MAHERY), Maroantsetra, Madagascar; [4]Department of Epidemiology, Harvard TH Chan School of Public Health, Boston, MA, USA; [5]Institut Halieutique et des Sciences Marines (IHSM), University of Toliara, Toliara, Madagascar; [6]Service de District de la Santé Publique, Ministère de la Santé Publique, Toliara, Madagascar; [7]Centre Hospitalier Universitaire des Soins et de Santé Publique Analakely (CHUSSPA), Service de la Formation et la Recherche (SFR), Antananarivo, Madagascar; [8]Department of Social Behavioral Sciences, Harvard TH Chan School of Public Health, Boston, MA, USA; [9]Department of Environmental Health, Harvard TH Chan School of Public Health, Boston, MA, USA and [10]Department of Global Health and Population, Harvard TH Chan School of Public Health, Boston, MA, USA

## Abstract

There have been no culturally validated measures to screen for depression in Madagascar. In 2022–2023, we conducted qualitative studies in the Bay of Ranobe area in southwestern Madagascar to understand local mental health syndromes specific to this region. We found that the 8-item Patient Health Questionnaire (PHQ-8) shares symptoms with the general distress-like, depressive-like and grief-like syndromes elicited locally. We adapted the PHQ-8 to align with the unique symptoms found in the region that were missing from the measure. We administered the adapted PHQ-8 to 809 participants aged 16 and above. We found that the one-factor (Depression) model (root mean square error of approximation [RMSEA] = 0.046, standardized root mean square residual [SRMR] = 0.053, Comparative Fit Index [CFI] = 0.993 and Tucker–Lewis Index [TLI] = 0.991) had a better fit to our data than the two-factor (Cognitive–Affective and Somatic) model (RMSEA = 0.047, SRMR = 0.052, CFI = 0.994 and TLI = 0.990). The one-factor (Depression) model demonstrated good internal consistency (MacDonald's omega coefficient $\omega_0 = 0.81$ and ordinal alpha $\alpha_0 = 0.87$). We conducted a multigroup confirmatory factor analysis to establish measurement invariance (MI) across four groups (sex, ethnicity, level of education and age group) and found that all levels of MI were achieved across groups. Our research provides a validated method to assess the probable prevalence of current depression in southwestern Madagascar.

## Impact statement

The southwestern region of Madagascar has one psychiatrist for roughly 1.8 million people. This study assessed the 8-item Patient Health Questionnaire (PHQ-8) as the first validated measure to screen for depression in Madagascar. We conducted a rigorous process to adapt and validate the PHQ-8, using qualitative methods to culturally contextualize the measure and quantitative methods to assess its psychometric properties. Instead of translating depressive symptoms from English to the local dialects, we used the local idioms and vernacular from the qualitative studies to describe symptoms in the adapted PHQ-8. This culturally validated version of the PHQ-8 can help estimate the probable prevalence of depression in that region, where mental disorder data are scarce. The presence of this tool will be broadly useful to governmental, nongovernmental, and all relevant public health and aid organizations. Nonmental health specialists can administer it to screen for depression, and they can refer a patient with high depressive symptoms to the psychiatry unit at Centre Hospitalier Universitaire (University Hospital Center) in Toliara. The adapted PHQ-8 can serve as a monitoring and evaluation screening tool during mental health interventions in southwestern Madagascar. The adapted PHQ-8 can also be used during post-disaster recovery responses. Extreme weather events, such as cyclones, can be traumatic and are becoming increasingly frequent in Madagascar. Thus, the PHQ-8 can be used to identify people emotionally affected by cyclones and in need of support. Broadly, given that Western measures





may not adequately translate into low- and middle-income (LMIC) settings, our methodology can be used to culturally adapt measures in other similar regions.

## Introduction

Depression is the second most common mental disorder globally, with an estimated 280 million people suffering from the disorder in 2019 (World Health Organization [WHO], 2022). Globally, the 8-item Patient Health Questionnaire (PHQ-8) has been used to measure the prevalence of depression and depressive symptoms in large population-based studies in Europe, Africa and the United States (Kroenke et al., 2009; Dhingra et al., 2011; Arias-de la Torre et al., 2021; Osborn et al., 2022; Arias-de la Torre et al., 2023). The PHQ-8 is a brief screening measure for depressive symptoms, comprising the eight-item diagnostic criteria from the Diagnostic and Statistical Manual of Mental Disorders, 4th edition (DSM-IV) (Kroenke and Spitzer, 2002). Unlike the 9-item Patient Health Questionnaire (PHQ-9), which has been widely validated in 10 Sub-Saharan African countries (Carroll et al., 2020), the PHQ-8 has been validated in only two African countries: Nigeria and Kenya (Aloba et al., 2018; Osborn et al., 2022). Neither the PHQ-9 nor the PHQ-8 has been used in any psychiatric epidemiological studies in Madagascar.

To date, there are currently no adapted and validated measures to screen for depression in Madagascar. Thus, there have been no studies or data that representatively assessed the prevalence of depression or depressive symptoms in Madagascar at local, regional or national levels. Only one epidemiological study has been conducted using a nonvalidated measure to screen for depression and assess the association between depressive symptoms, socioeconomic status and major life stressors among adults recruited from nonclinical settings in northern Madagascar (Foubert et al., 2021). Comprehending and eliciting local concepts of mental disorders are crucial steps in adapting existing measures to local contexts (Bass et al., 2007). Only a few studies have considered this approach when adapting or developing measures to screen for depression in Africa (e.g., Bolton, 2001; Bolton et al., 2004; Betancourt et al., 2009), and there is now a greater emphasis on incorporating culturally specific symptoms into Western measures for global mental health research in LMICs. In this study, we used qualitative data to understand the local conceptualization of mental health syndromes in southwestern Madagascar and evaluated the psychometric properties of the adapted PHQ-8 in this context.

Madagascar is one of the world's poorest countries, with ~80.7% of its 30.3 million people living under the poverty threshold (World Bank, 2024). The country also faces a number of public health challenges, having one of the world's highest global hunger indices of 36.3, which falls into the alarming category, a very high prevalence of stunting (39.8%) (Global Hunger Index, 2024) and a high prevalence of micronutrient deficiencies (Golden et al., 2024a). Madagascar lacks mental healthcare specialists, with only 24 psychiatrists, or <1 psychiatrist per 1,000,000 people. Poverty and food insecurity are often associated with poor mental health, which may lead Malagasy people to be more vulnerable to mental disorders (Lund et al., 2010, 2018; Pourmotabbed et al., 2020; Ridley et al., 2020; Trudell et al., 2021; Kirkbride et al., 2024). Compounding challenges with food systems, poverty and malnutrition, Madagascar also faces the detrimental impacts of climate change, such as drought, cyclones and floods, which have been associated with mental health (Berry et al., 2010; Charlson et al., 2021; World Bank, 2021; Burrows et al., 2024; Rigden et al., 2024; Hadfield et al., 2024).

The syndemic of climate change, extreme weather events and food system failures may adversely impact the mental health of the Madagascar population. Thus, a culturally validated measure to screen for depression, such as the PHQ-8, is crucial to assess the impact of poverty, food insecurity and climate change on mental health.

The main objectives of this article are to (1) describe our culturally informed adaptation of the PHQ-8 to the Malagasy population in southwestern Madagascar; (2) assess the reliability, factor structure and measurement invariance (MI) of the PHQ-8 and (3) estimate the probable prevalence of current depression in southwestern Madagascar.

## Methods

### Qualitative study

#### Study participants and procedures

In 2022 and 2023, we collected qualitative data (6 focus group discussions [FGDs], 32 free listing [FL] interviews and 23 cognitive interviews with key informants [KI] see subsequent sections) to elicit local mental health syndromes in the Bay of Ranobe (BoR) (Figure 1) with their associated causes, coping strategies and symptoms by using components of the Design, Implementation, Monitoring and Evaluation (DIME) process modules (Bolton and Tang, 2004; Applied Mental Health Research Group, 2013). The main goal was to adapt an existing depression measure that can match depressive-like syndromes in the BoR with their associated symptoms. Unlike the DIME process, we started with FGDs to generate concepts of all local mental health syndromes. Then, we followed the first three modules of the DIME process: (1) conduct a qualitative assessment, (2) develop/adapt the measure and (3) assess the probable prevalence of current depression at baseline.

#### Focus group discussion

We conducted six FGDs to list local mental health syndromes in four communities, two coastal and two inland, in the BoR area. A total of 48 individuals participated in the FGDs. The participants included male and female adolescents and adults. We purposely included adolescents and adults because variations of local syndromes, vernacular and idioms might differ across age groups. The ages of the adolescent participants ranged from 16 to 22 years, whereas the ages of the adult participants were above 22 years. Each FGD was composed of six participants and was conducted separately for males and females. The FGD length ranged from 45 min to 1 h. We used convenience sampling to recruit the participants. FGDs were conducted by two people. One person led the discussion, while the other served as a scribe. The FGD team consisted of three physicians with mental health specialization, one public nurse and three researchers, including some of the coauthors (CDG, AFM, NVV, HOR and HJR). A mix of Masikoro, Vezo and Merina dialects was used when conducting the FGDs. We asked for verbal consent before starting the FGDs, following Harvard University Institutional Review Board (IRB) and locally approved protocols. First, we broadly inquired about the major problems faced by the communities to see if any psychosocial problems would emerge from the FGDs and used probing to elicit responses. If any local

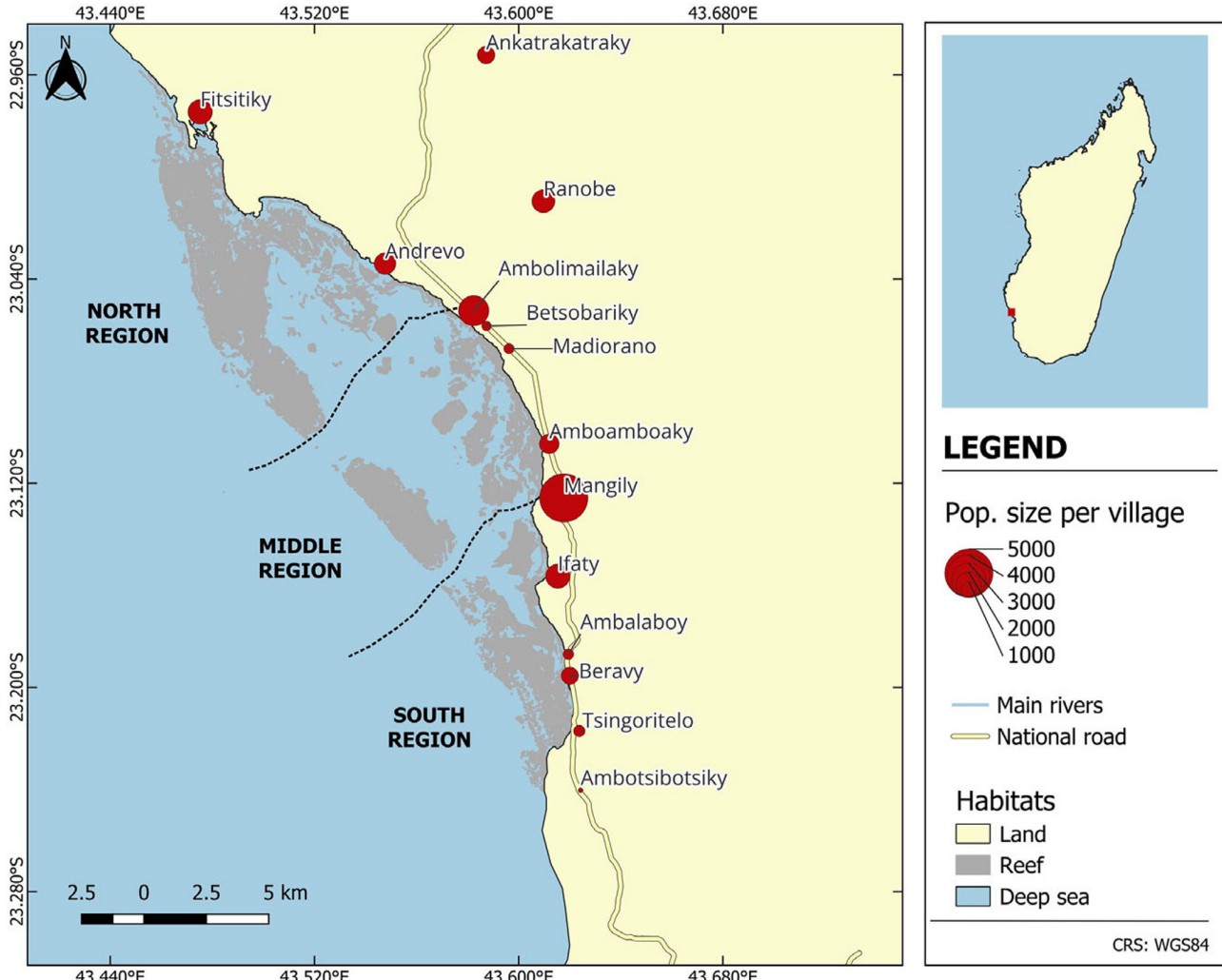

**Figure 1.** Study sites of the HIARA cohort in the Bay of Ranobe, southwestern Madagascar.

syndromes emerged from the psychosocial problems, we probed the participants on the causes, coping strategies and signs/symptoms. At the end of the FGD, we provided $10USD for each group to compensate for their time. We also asked the participants to identify people with whom they consulted when experiencing these local mental health syndromes. A debriefing was conducted at the end of the day to review all notes and discuss any local syndromes that frequently emerged from the FGDs. HJR translated the team's notes from the Vezo and Masikoro dialects into English.

*Free listing interviews*
After generating a list of common local mental health syndromes from the FDGs, we conducted 32 FL interviews to elicit the causes, coping strategies and symptoms associated with these syndromes. We conducted a 1-week training session for interviewers on qualitative methodology, mainly focused on the FL techniques. The FL interviews were administered individually in the Vezo and Masikoro dialects. These two dialects are part of the southwestern group dialects and have distinctive features (e.g., the use of the glottal stop, the adoption of the demonstrative pronoun *i*, the use of *te* in lieu of *ŋe* or *ni* and the existence of many roots ending in *-kē*) compared to other Malagasy dialects (Adelaar, 2013; Serva and Pasquini, 2020). Two local university

students (R and MR), who are both fluent in the Vezo and Masikoro dialects, conducted the FL interviews under the supervision of HJR, who is a native Malagasy speaker. The interviewer probed the participants to briefly describe their answers to each question. A note-taker recorded all responses to the survey questionnaire. We used a convenience sample, and participants (15 females and 17 males) were sampled from 7 communities and a diverse range of occupations (e.g., fishers, fishmongers, farmers, faith leaders and unemployed) living in the BoR. The median age of the participants was 49 years (range: 18–75 years). We asked for verbal consent before each FL and provided $1USD per participant to compensate for the <1 h spent on the interview. Each FL lasted, on average, 45 min (SD = 1 min). For data analysis, R and MR first cleaned the notes before proceeding to the FL analysis. We used content and thematic analysis. First, R, MR and HJR reviewed and discussed a list of brief descriptions of each response question. Second, similar short descriptions were grouped into categories. Third, the numbers of brief descriptions for each category were summed to get a frequency of the category. Fourth, categories were sorted by frequency, from highest to the lowest. The content and thematic analyses were conducted before the English translation. HJR translated the FL analysis into English.

### Cognitive interviews with key informants

We translated the PHQ-8 into Vezo and Masikoro dialects using the wording, idioms and vernacular from the FL interviews. We then mapped the local syndromes from the FGDs and FLs to the DSM 5th edition text revision (DSM-5-TR) symptoms of depression and the translated version of the selected depression measure (PHQ-8) to examine the overlap. We then conducted 23 cognitive interviews with KIs (14 females and 9 males) to understand how the adapted PHQ-8 items were understood in the local community. Each cognitive interview lasted, on average, 43 min (SD = 1 min). Specifically, we reviewed each item of the PHQ-8 using the exact wording we obtained during the FL interviews. We also asked the KIs about the symptoms associated with the three local mental health syndromes, but not in-depth as in the FL interviews. For each PHQ-8 item, we asked them how they understood the wordings and their meaning. If some words and phrases were ambiguous or unclear, we asked for suggestions to improve their clarity. The KIs comprised primary mental healthcare providers mentioned by the FL participants, such as traditional healers (herbalists, mediums and diviners), Christian faith healers, community health volunteers, traditional midwives and nurses. HJR and NVV conducted the cognitive interviews with KIs. We asked for verbal consent, audio-recorded each interview, and did not mention the name of the KI in the recordings. Each KI received \$2USD to compensate for the <1 h spent on the interview. NVV completed the transcription of the audio recordings, and two local people translated the audio transcription into English. Similar to the FL analysis, we conducted content and thematic analysis of the cognitive interviews with KIs. HJR reviewed a list of brief descriptions of each response question in the Vezo and Masikoro dialects. Second, similar brief descriptions were grouped into categories. Third, the numbers of brief descriptions for each category were summed to get a frequency of the category. Fourth, categories were sorted by frequency. The content and thematic analyses were conducted before the English translation. HJR translated the cognitive interview analysis into English.

### Quantitative study

#### Study participants and procedures

The study participants above the age of 16 years ($N$ = 809) for the psychometric analysis are part of the ongoing Health Impacts of Artificial Reef Advancement (HIARA) cohort study in southwestern Madagascar (Golden et al., 2024b). The BoR (Figure 1) is a biodiversity hotspot with a 32-km-long coral reef barrier. Coral reefs have been experiencing environmental degradation, including unsustainable fishing practices and coral bleaching. One of the primary aims of the HIARA cohort is to restore coral reefs, rebuild fisheries and improve the health and well-being of the BoR communities (Golden et al., 2024b). Our team was interested in mental health in this area specifically, as we had selected the area for a longitudinal cohort study to evaluate the effects of environmental change on both aquatic and terrestrial food systems. Given the existing partnerships and infrastructure for the longitudinal study, as well as the existing qualitative observations of climate and environmental change, it was an ideal location for this study. Although the HIARA cohort study began in January 2023, the mental health module of the HIARA cohort study has been administered to individuals aged 16 years and above since October 2023. FD, MYS, JCM, R and MR collect measurements for the same participants every 3 months (R and MR also conducted the FL interviews). The PHQ-8 was administered individually to each participant. No household members from the qualitative study were randomly selected to participate in the HIARA cohort study. We used the data collected in October 2023 for the psychometric analysis. The HIARA cohort study enrolled a total of 1,539 participants from 12 communities that reside along the BoR, where fishing-related activities are the primary source of livelihood, and 2 inland communities adjacent to the BoR, where agriculture is the primary source of livelihood.

From each of the 12 coastal communities in the BoR, 30 households were randomly sampled across four categories: (1) households with at least one individual engaged in fishing activities and at least one child under 5 years of age, (2) households with at least one fisher but no children under 5 years of age, (3) households with at least one child under 5 years of age but no fisher and (4) households with neither children under 5 years of age nor fishers. In contrast, for the 2 inland communities, we randomly sampled 45 households in each community across two categories: (1) households with at least one farmer and (2) households without farmers (Golden et al., 2024b). The aforementioned study participants ($N$ = 809) enrolled in the mental health module of the study all belong to these 450 households, and all participants verbally consented to participate in the study following Harvard University IRB and locally approved protocols (Golden et al., 2024b).

### Measure

*The PHQ-8.* The PHQ-8 is a Likert-type self-reported measure to screen for depression. The eight items are based on the DSM-IV (American Psychiatric Association, 1994) criteria for major depression (anhedonia, depressed mood, sleep disturbance, fatigue, appetite changes, low self-esteem, concentration difficulties and psychomotor disturbances). We asked the number of days in the past 2 weeks (14 days) the participants had experienced each of the eight items: 0–1 day ("Not at all"), 2–6 days ("Several days"), 7–11 days ("More than half the days") and 12–14 days ("Nearly every day") (Dhingra et al., 2011). After summing scores across all eight questions, total scores were classified as no significant symptoms (0–4), mild symptoms (5–9), moderate symptoms (10–14), moderately severe symptoms (15–19) and severe symptoms (20–24) (Kroenke and Spitzer, 2002; Kroenke et al., 2009). The PHQ-8, which excludes the suicidal ideation item from the PHQ-9, has been shown to perform as well as the PHQ-9 in predicting probable current depression (Kroenke et al., 2009).

### Statistical approach

We first estimated the probable prevalence of current depression of the study participants ($N$ = 809) in October 2023 using the PHQ-8 cutoff score of 10 (Kroenke et al., 2009) across groups (sex, age group, marital status and area). We used a $\chi^2$-test to compare the probable prevalence of current depression across groups. We used the "lavaan" (Version 0.6-17) and "semTools" (Version 0.5-6) packages in RStudio (Version 2024.09.0+375) for the analysis (Rosseel, 2012; RStudio Team, 2020; Jorgensen et al., 2022).

### Confirmatory factor analysis

For the subsequent psychometric analysis of the PHQ-8, we treated the data as ordinal indicators. Therefore, we used polychoric correlations to estimate the association between the continuous latent response variables, which reflect the ordinal observed indicators. We used delta scaling parameterization by fixing the variance of the common factors to 1 to estimate the parameters. We used the diagonal weighted least squares (DWLS) estimator method, which

is appropriate for ordinal data with fewer than five categories (Kline, 2023).

### Model fit

We conducted a confirmatory factor analysis (CFA) to evaluate whether a one-factor (Depression) model or a two-factor (Somatic and Cognitive–Affective) model of the PHQ-8 better fit our data. Specifically, for the two-factor model, items 3 (Sleep disturbance), 4 (Fatigue) and 5 (Appetite changes) loaded on the Somatic factor, whereas items 1 (Anhedonia), 2 (Depressed mood), 6 (Low self-esteem), 7 (Concentration difficulties) and 8 (Psychomotor disturbance) loaded on the Cognitive–Affective factor (Lamela et al., 2020). To evaluate the model fit, we conducted a separate $\chi^2$-test for each model. Because the $\chi^2$-test is sensitive to large sample sizes, we considered other model fit indices, such as the root mean square error of approximation (RMSEA), the standardized root mean square residual (SRMR), the Bentler Comparative Fit Index (CFI) and the Tucker–Lewis Index (TLI). We used a cutoff criterion value ≥0.95 for the CFI and TLI as criteria of goodness of fit. For RMSEA and SRMR, the criteria for goodness of fit are ≤0.08 (Hu and Bentler, 1999; Kline, 2023). We conducted a $\chi^2$-difference test (one-factor vs. two-factor model) to decide which model we retained for the multigroup CFA (MG-CFA). The best model (one-factor vs. two-factor models) we retained was the baseline model for measurement invariance (MI) analysis.

### Reliability and convergent validity

We computed the McDonald's omega coefficient ($\omega$) as a measure of composite reliability, evaluated the standardized factor loadings and assessed the average variance extracted (AVE) to establish the convergent validity of the two models. In addition, we followed the recent recommendations of Cheung et al. (2023) on convergent validity assessment by considering sampling errors. That is, an omega coefficient ($\omega$) above 0.7, a standardized factor loading with a 90% upper limit confidence interval (ULCI) above 0.5 and an AVE with a 90% ULCI above 0.5 are evidence of convergent validity (Cheung et al., 2023). We also computed the ordinal coefficient alpha (α) as another measure of reliability instead of the traditional Cronbach's alpha, given that the latter has been found to underestimate the reliability coefficient for ordinal item data (Zumbo et al., 2007; Gadermann et al., 2012).

### Measurement invariance across groups

We conducted an MG-CFA analysis across groups (i.e., sex, ethnicity, level of education and age category) to evaluate MI. In the MG-CFA analysis, for the level of education group, we collapsed secondary (6–9 years), high school (10–12 years) and higher education (13+ years) into one category and dropped observations in which the education attainment data were unavailable. Thus, we had no education level, primary education level (1–5 years) and secondary education and above level (6+ years) for the MG-CFA. Similarly, for the age group, we combined participants aged 45–59 years and above 60 years. This was done to have at least 100 participants per group. Thus, we had three age groups: 16–26 years, 30–44 years and 45+ years. We used Wu and Eastbrook's procedure (2016) to assess the MI by evaluating configural invariance, thresholds invariance and thresholds and loadings invariance, which were operationalized by Svetina et al. (2020; Wu and Estabrook, 2016). Specifically, we first assessed the configural invariance to ascertain if the construct has the same pattern of factor loadings across groups. Second, we constrained the thresholds to be the same across groups. Third, we both constrained the thresholds and

loadings to be the same across groups. To evaluate MI, we used a sequential $\chi^2$-difference test to compare nested models. Indeed, the thresholds invariance model is nested in the configural invariance model, and the thresholds and loadings invariance model is nested in the thresholds invariance model. In addition to the $\chi^2$-test difference, which is very sensitive to the sample size, we used Chen's (2007) cutoff criteria to test for MI. That is, an absolute change in CFI ($\Delta$CFI $\leq -0.010$) and RMSEA ($\Delta$RMSEA $\leq 0.015$) indicates MI (Chen, 2007).

## Results

### Qualitative study

#### Focus group discussions

We found three common local syndromes (Figure 2) that are similar to general distress-like, depressive-like and grief-like syndromes: *Fiasan-doha* (head working), *Alahelo maré* (deep sadness) and *Jangobo maré* (deeply missing someone). We used these local mental health syndromes from the FGDs in FL and cognitive interviews with KIs.

#### Free listing interviews

Table 1 indicates the symptoms found in the three local mental health syndromes and the diagnostic criteria for major depressive disorder in the DSM-5-TR (American Psychiatric Association, 2022). Most of the symptoms are found in the DSM-5-TR. However, there are many symptoms specific to the local mental health syndromes not found in the DSM-5-TR (Figure 2). *Fiasan-doha* is similar to a general distress-like syndrome and shares symptoms of the "Thinking too much" syndrome and idiom that includes both mood and anxiety disorder symptoms (irritability, headache and easily startled) (Kaiser et al., 2015). *Alahelo maré* is a depressive-like and grief-like syndrome that shares some of the symptoms found in the DSM-5-TR to diagnose major depression, such as unmotivated (anhedonia), depressed mood, feeling weak, sleeping and suicidal thoughts. Similarly, *Jangobo maré* is also a depressive-like and grief-like syndrome mainly caused by the end of a romantic relationship. *Jangobo maré* also shares some of the symptoms found in the DSM-5-TR to diagnose major depression, such as anhedonia (unmotivated), depressed mood, feeling weak, sleeping and suicidal thoughts. Both *Alahelo maré* and *Jangobo maré* included possible psychotic symptoms, such as "speak nonsense," "become crazy" and "self-talk" (Figure 2). There were nine symptoms (Figure 2) that overlapped for *Alahelo maré* and *Jangobo maré* syndromes. Interestingly, four symptoms (unmotivated, depressed mood, irritability and losing weight) overlapped for the three syndromes, and these symptoms included the two main symptoms that must be present to be clinically diagnosed with depression: unmotivated (anhedonia) and depressed mood (Figure 2).

A clinical psychologist (KCK) assessed the content validity of the adapted measure for depression after reviewing the three local syndromes and their translated associated symptoms in English. Based on the FL analysis, the original PHQ-8 was adapted to the Malagasy population in the BoR in southwestern Madagascar.

#### Cognitive interviews with key informants

Some items needed to be modified due to the vagueness of the local word. For instance, the word *Tsy Mazoto* (unmotivated), a symptom mostly reported by the participants found in the three local mental health syndromes as a sign of lack of interest or anhedonia, can be interpreted as laziness, boredom and tiredness without

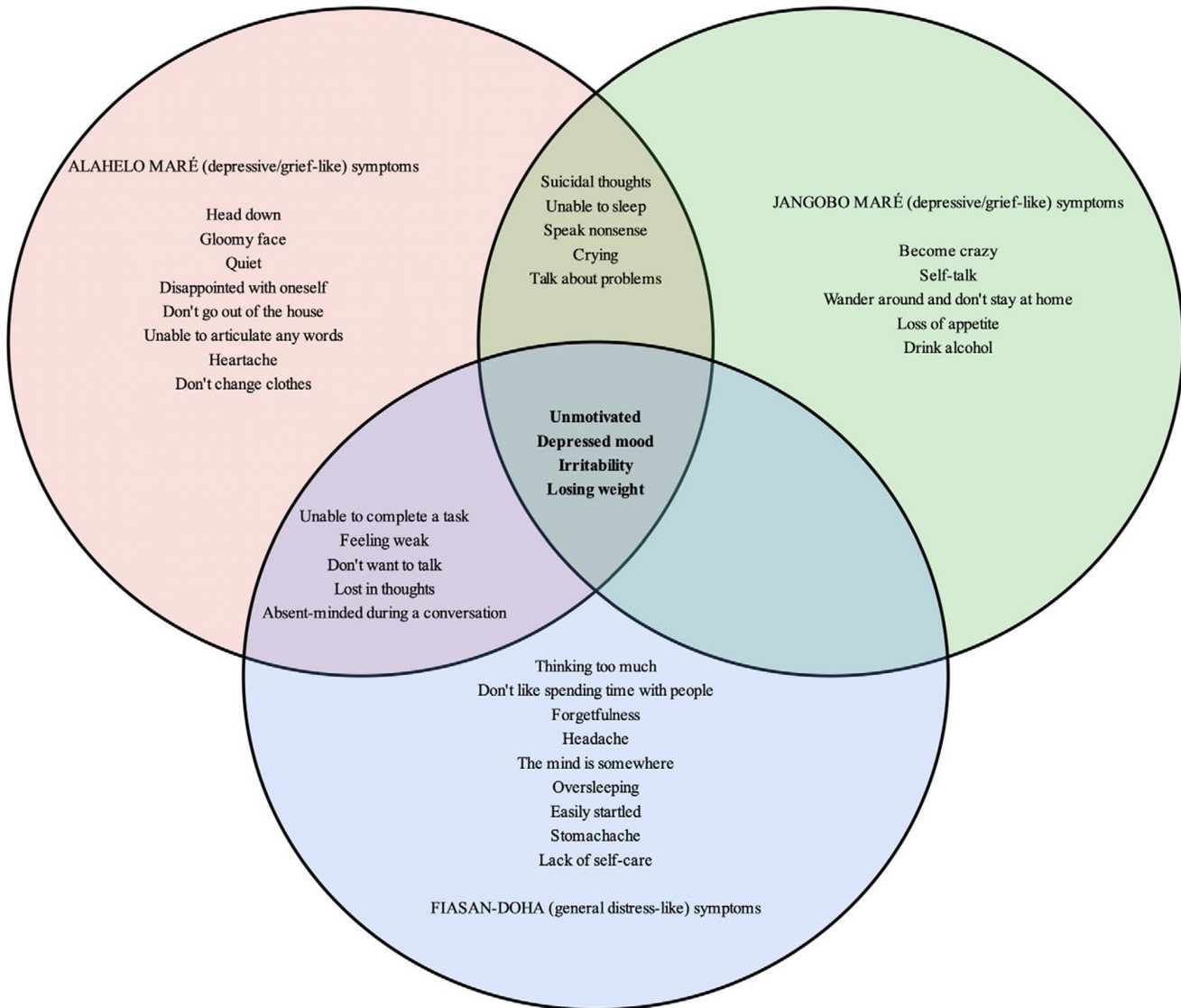

**Figure 2.** Local mental health syndromes (*Fiasan-doha, Alahelo maré* and *Jagombo maré*) and their associated symptoms.
*Note*: The frequency of these symptoms was combined for the free listing and cognitive interviews with KIs. Only symptoms that were reported at least two times are kept in this figure, except for suicidal thoughts, which were reported only once but added for their relevance.

contextualization during the cognitive interviews with KI. Thus, to avoid confusion, we had to explicitly include and translate the phrase "little interest and pleasure" in the original item 1 of PHQ-8 without using the words *Tsy Mazoto*.

After finalizing the PHQ-8, we back-translated the PHQ-8 into English. There are no major differences between the back translation of the adapted PHQ-8 and the original PHQ-8.

### Quantitative study

#### Probable prevalence of current depression
Our survey comprised a roughly even mix of individuals by sex (58.8% female), ethnicity (roughly half Vezo, and an even mix of Masikoro, Antandroy and other ethnic mixes) and education (one-quarter of the population with no education, one-third with primary education and one-quarter with secondary and above education). The average age of participants was 36.9 years, and more than

two-thirds of the population were married. Approximately 8% (95% confidence interval [CI]: 6.30–10.18%) of the participants had a PHQ-8 score of 10 or above, which indicated a probable current depression (Table 2). The probable prevalence of current depression in males (6%) and females (9.5%) was not statistically significantly different (Figure 3A; $\chi^2 = 2.7$, df = 1, $p = 0.10$). Among the age groups, the probable prevalence of current depression increased with age: 16–29 years (3.6%), 30–44 years (8%), 45–59 years (10.6%) and 60+ (21.4%) years. There was an overall statistically significant difference in prevalence across age groups (Figure 3B; $\chi^2 = 30.54$, df = 3, $p < 0.001$). The probable prevalence of current depression in the coastal area (9.5%) was statistically significantly higher than in the inland area (1.5%) (Figure 3C; $\chi^2 = 8.16$, df = 1, $p = 0.004$). Among marital status groups, there was an overall statistically significant difference in the probable prevalence of current depression (Figure 3D; $\chi^2 = 41.25$, df = 3, $p < 0.001$) and widowed (40%) had the highest prevalence.

**Table 1.** Local mental health syndromes and DSM-5-TR symptoms for major depressive disorder

| Major depressive disorder symptoms (DSM-5-TR) | Fiasan-doha | Alahelo maré | Jangobo maré |
|---|---|---|---|
| Depressed mood | X | X | X |
| Diminished interest or pleasure in activities | X | X | X |
| Weight loss | X | | X |
| Weight gain | | | |
| Decrease in appetite | | X | X |
| Increase in appetite | | | |
| Insomnia | | | X |
| Hypersomnia | X | | |
| Psychomotor agitation | | | X |
| Psychomotor retardation | | X | |
| Fatigue or loss of energy | X | X | |
| Worthlessness or guilt | | X | |
| Unable to concentrate | X | | |
| Thoughts of death or suicidal ideation | | X | X |

### Confirmatory factor analysis

The estimated standardized factor loadings represented the correlation between common factors and the theoretical continuous latent response variables: PHQ-1* (Anhedonia), PHQ-2* (Depressed mood), PHQ-3* (Sleep disturbance), PHQ-4* (Fatigue), PHQ-5* (Appetite changes), PHQ-6* (Low self-esteem), PHQ-7* (Concentration difficulties) and PHQ-8* (Psychomotor disturbance; see Figure 4).

### Model fit

We found that both the one-factor (Depression) model (RMSEA = 0.046, SRMR = 0.053, CFI = 0.993 and TLI = 0.991) and the two-factor (Cognitive–Affective and Somatic) model (RMSEA = 0.047, SRMR = 0.052, CFI = 0.993 and TLI = 0.990) had a good model fit (Table 3). However, the $\chi^2$-difference test ($p = 0.208$) concluded that the one-factor model had a better fit and was more parsimonious than the two-factor model (Table 3).

### Reliability and convergent validity

We also found that the convergent validity of the one-factor (Depression) model was established. The omega coefficient ($\omega_0$) was 0.81, and the ordinal alpha ($\alpha_0$) was 0.87 (Table 4). PHQ-4* (Fatigue) had the lowest standardized factor loading (0.486), whereas PHQ-1* (Anhedonia) had the highest standardized factor loading (0.791) (Figure 4A). All loadings had a 90% ULCI > 0.5 (Table 4). Overall, the depression factor explained, on average, 48.50% (AVE = 0.485; 90% ULCI > 0.5) of the variance of eight continuous latent response variables (Table 4).

Conversely, the convergent validity of the two-factor (Cognitive–Affective and Somatic) model was not supported. For the cognitive-affective factor, the omega coefficient subscale ($\omega_1$) was 0.78, and the ordinal alpha ($\alpha_1$) was 0.88 (Table 4). The standardized factor loadings ranged from 0.738 to 0.793 (Figure 4B). All standardized factor loadings had a 90% ULCI above 0.7 (Table 4). The cognitive-affective factor, on average, accounted for 58.81% (AVE = 0.588; 90% ULCI > 0.5) of the variance of the PHQ-1* (Anhedonia), PHQ-2* (Depressed mood), PHQ-6* (Low self-esteem), PHQ-7*

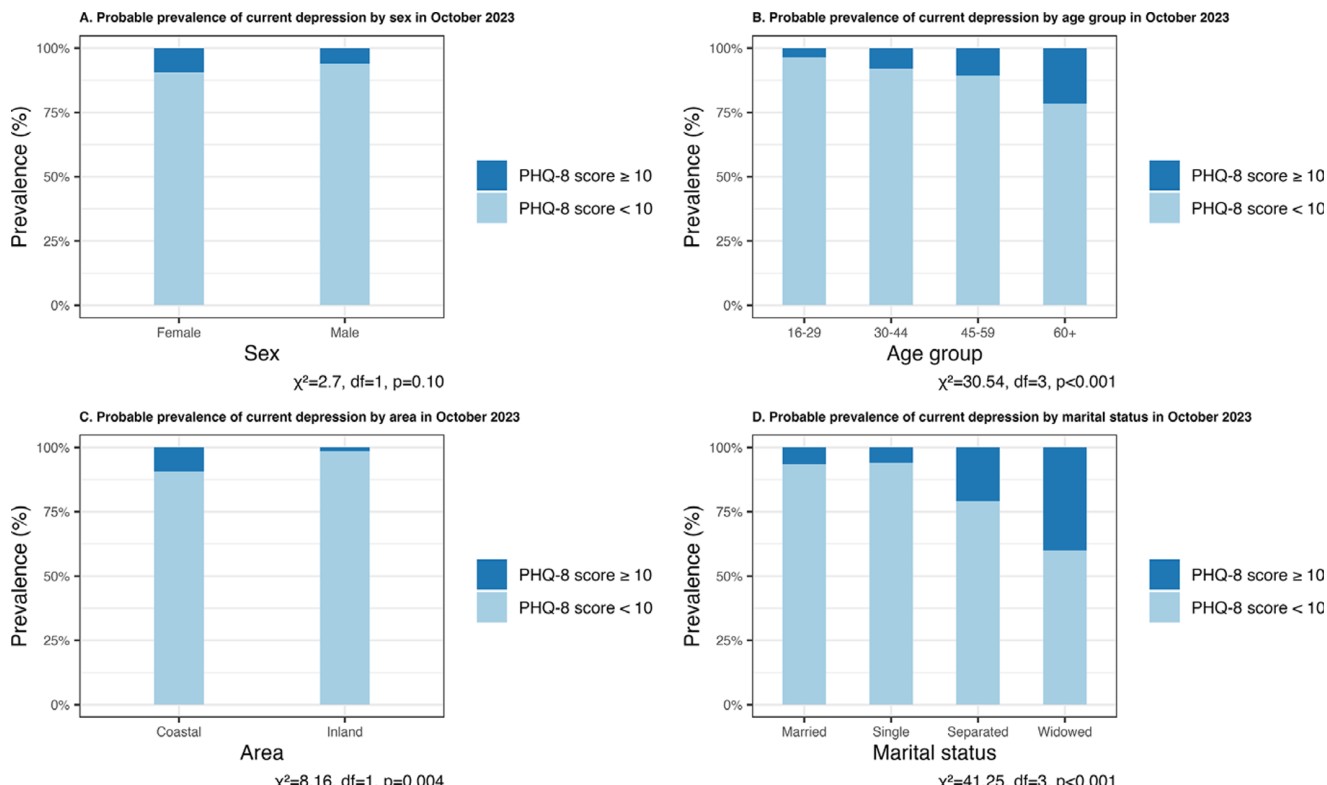

**Figure 3.** Probable prevalence of current depression among adults above 16+ by sex, age group, marital status and area in October 2023 in the HIARA cohort study.

**Table 2.** Study participant characteristics (*N* = 809)

| | *N* (%) |
|---|---|
| Full sample | 809 (100%) |
| Sex | |
| Male | 476 (58.8%) |
| Female | 333 (41.2%) |
| Age (years) | |
| Mean (SD) | 36.9 (16.1) |
| Age group | |
| 16–29 | 333 (41.2%) |
| 30–44 | 251 (31.0%) |
| 45–59 | 141 (17.4%) |
| 60+ | 84 (10.4%) |
| Marital status | |
| Married | 553 (68.4%) |
| Single | 188 (23.2%) |
| Separated | 48 (5.9%) |
| Widowed | 20 (2.5%) |
| Level of education | |
| No education | 203 (25.1%) |
| Primary education (1–5 years) | 263 (32.5%) |
| Secondary education (6–9 years) | 181 (22.4%) |
| High school (10–12 years) | 28 (3.5%) |
| Higher education (13+ years) | 60 (7.4%) |
| Not applicable | 74 (9.1%) |
| Ethnicity | |
| Vezo | 408 (50.4%) |
| Masikoro | 160 (19.8%) |
| Antandroy | 123 (15.2%) |
| Other/Mixed | 118 (14.6%) |
| Depressive symptoms | |
| No symptoms (0–4) | 357 (44.1%) |
| Mild (5–9) | 387 (47.8%) |
| Moderate (10–14) | 57 (7.0%) |
| Moderately severe (15–19) | 3 (0.4%) |
| Severe (20–24) | 5 (0.6%) |
| PHQ–8 score | |
| PHQ–8 score ≥ 10 | 65 (8.0%) |
| PHQ–8 score <10 | 744 (92.0%) |

(Concentration difficulties) and PHQ-8* (Psychomotor disturbance). For the somatic factor, the omega coefficient subscale ($\omega_2$) was 0.52, and the ordinal alpha ($\alpha_2$) was 0.58. The standardized factor loadings ranged from 0.501 to 0.663 (Figure 4B). The somatic factor, on average, accounted for only 34.21% (AVE = 0.394; 90% ULCI < 0.5) of the variance of the PHQ-3* (Sleep disturbance), PHQ-4* (Fatigue) and PHQ-5* (Appetite changes), and these three items had the lowest loadings (Table 4). The somatic and cognitive-affective factors were highly correlated (0.949) (Figure 4).

### *Measurement invariance across groups*

We found full measurement invariance (configural, thresholds and thresholds and loadings invariances) across groups (i.e., sex, ethnicity, level of education and age group) for the one-factor (Depression) model (Table 5). For the configural invariance, the models for sex (RMSEA = 0.070, CFI = 0.975 and TLI = 0.965), ethnicity (RMSEA = 0.060, CFI = 0.982 and TLI = 0.974); level of education (RMSEA = 0.085, CFI =0.969 and TLI = 0.948) and age groups (RMSEA = 0.074, CFI = 0.969 and TLI = 0.957) provided an acceptable fit (Table 5). Configural invariance was established, which indicates that depression has the same pattern of factor loadings for each group. Thresholds invariance for sex ($\Delta$ RMSEA = −0.005 and $\Delta$ CFI = −0.002), ethnicity ($\Delta$ RMSEA = 0.000 and $\Delta$ CFI = −0.005), level of education ($\Delta$ RMSEA = −0.007 and $\Delta$ CFI = −0.002) and age group ($\Delta$ RMSEA = −0.001 and $\Delta$ CFI = 0.006) were also achieved. Thresholds and loadings invariances were also supported for sex ($\Delta$ RMSEA = −0.006 and $\Delta$ CFI = 0.002), ethnicity ($\Delta$ RMSEA = −0.011 and $\Delta$ CFI = 0.005), level of education ($\Delta$ RMSEA = −0.013 and $\Delta$ CFI = 0.007) and age groups ($\Delta$ RMSEA = −0.004 and $\Delta$ CFI = −0.002), which indicated that (1) the depression construct has the same meaning across groups, and (2) any group differences in the depression mean scores, latent response variables mean scores and observed ordinal mean scores are unbiased.

## Discussion

### *Fiasan-Doha, Alahelo Maré and Jangobo Maré syndromes*

The qualitative studies demonstrated that the local mental health syndromes (*Fiasan-doha*, *Alahelo maré* and *Jangobo maré*) have a shared construct similar to the depressive symptoms found in the DSM-5-TR for major depression. We aimed to use the exact wording we obtained from our qualitative studies to adapt the PHQ-8, ensuring that we adequately captured the emotions and feelings of these syndromes in southwestern Madagascar, as these can differ from culture to culture. Directly translating existing measures without considering local idioms, vernacular and cultural context can be misleading for this reason (Bass et al., 2007).

The results of our qualitative studies were also consistent with previous studies on how depression is expressed worldwide, with each local syndrome including the two main symptoms of

**Table 3.** Global model fit statistics of the one-factor (Depression) and two-factor (Cognitive–Affective and Somatic) PHQ-8 models

| Model | $\chi 2$ | df | p | RMSEA | 90% CI | SRMR | CFI | TLI | $\Delta\chi 2$ | p ($\Delta\chi 2$) |
|---|---|---|---|---|---|---|---|---|---|---|
| One-factor (Depression) model | 54.159 | 20 | <0.001 | 0.046 | [0.031–0.061] | 0.053 | 0.993 | 0.991 | | |
| Two-factor (Cognitive–Affective and Somatic) model | 52.580 | 19 | <0.001 | 0.047 | [0.032–0.062] | 0.052 | 0.993 | 0.990 | 1.579 | 0.208 |

*Note*: CFI, Comparative Fit Index; RMSEA, root mean square error approximation; SRMR, standardized root mean square residual; TLI, Tucker–Lewis Index.

**Table 4.** DWLS unstandardized and standardized factor loadings, omega coefficients, ordinal alphas and average extracted variance (AVE) for one-factor (Depression) and two-factor (Cognitive–Affective and Somatic) PHQ-8 models with ordinal indicators

| Parameter | Unstandardized | | Standardized | | |
|---|---|---|---|---|---|
| | Estimate | SE | Estimate | SE | 90% CI |
| *One-factor (Depression) model* | | | | | |
| Depression factor | | | | | |
| Depression →PHQ–1* (Anhedonia) | 1.000 | | 0.791 | 0.019 | [0.759–0.823] |
| Depression → PHQ–2* (Depressed mood) | 0.955 | 0.037 | 0.755 | 0.019 | [0.723–0.787] |
| Depression → PHQ–3* (Sleep disturbance) | 0.806 | 0.032 | 0.638 | 0.019 | [0.606–0.669] |
| Depression → PHQ–4* (Fatigue) | 0.614 | 0.031 | 0.486 | 0.021 | [0.452–0.520] |
| Depression → PHQ–5* (Appetite changes) | 0.709 | 0.033 | 0.561 | 0.022 | [0.525–0.596] |
| Depression → PHQ–6* (Low self-esteem) | 0.931 | 0.037 | 0.736 | 0.021 | [0.702–0.770] |
| Depression → PHQ–7* (Concentration difficulties) | 0.991 | 0.038 | 0.784 | 0.020 | [0.750–0.817] |
| Depression → PHQ–8* (Psychomotor disturbance) | 0.957 | 0.042 | 0.757 | 0.026 | [0.714–0.800] |
| Omega coefficient $(\omega_0)$=0.81; Ordinal alpha ($\alpha_0$)=0.87 | | | | | |
| AVE = 0.485 90% CI [0.439–0.534] | | | | | |
| *Two-factor (Cognitive–affective and somatic) model* | | | | | |
| Cognitive-Affective factor | | | | | |
| Cognitive-Affective →PHQ–1* (Anhedonia) | 1.000 | | 0.793 | 0.020 | [0.761–0.825] |
| Cognitive-Affective →PHQ–2* (Depressed mood) | 0.955 | 0.037 | 0.757 | 0.020 | [0.725–0.789] |
| Cognitive-Affective →PHQ–6* (Low self-esteem) | 0.930 | 0.037 | 0.738 | 0.021 | [0.704–0.772] |
| Cognitive-Affective →PHQ–7* (Concentration difficulties) | 0.991 | 0.038 | 0.786 | 0.020 | [0.752–0.819] |
| Cognitive-Affective →PHQ–8* (Psychomotor disturbance) | 0.958 | 0.042 | 0.759 | 0.026 | [0.716–0.802] |
| Omega coefficient subscale $(\omega_1)$=0.78; Ordinal alpha subscale $(\alpha_1)$=0.88 | | | | | |
| AVE = 0.588 90% CI [0.536–0.643] | | | | | |
| Somatic factor | | | | | |
| Somatic → PHQ–3* (Sleep disturbance) | 1.000 | | 0.663 | 0.028 | [0.618–0.709] |
| Somatic → PHQ–4* (Fatigue) | 0.756 | 0.041 | 0.501 | 0.024 | [0.462–0.541] |
| Somatic → PHQ–5* (Appetite changes) | 0.857 | 0.044 | 0.579 | 0.027 | [0.534–0.623] |
| Omega coefficient subscale $(\omega_2)$=0.52; Ordinal alpha subscale $(\alpha_2)$=0.58 | | | | | |
| AVE = 0.342 90% CI [0.294–0.394] | | | | | |
| *Covariance* | | | | | |
| Cognitive–Affective and Somatic | 0.499 | 0.019 | 0.949 | 0.039 | [0.885–1.013] |

depression: unmotivated (anhedonia) and depressed mood in DSM-5-TR (Haroz et al., 2017; Viduani et al., 2024). However, psychotic features, such as "self-talk," "speak nonsense" and "become crazy" (Figure 2) were present in *Alahelo maré* and *Jangobo maré* syndromes. These psychotic features were also found in the DSM-5-TR as specifiers for the major depressive disorder diagnosis.

We also found that there are culturally specific symptoms that should be added to the original PHQ-8. These symptoms were associated with the local mental health syndromes (Figure 2). For example, crying, irritability and social isolation were prevalent and

associated with the local syndromes. These culturally specific symptoms are consistent with other studies on how depression is expressed in different cultures (Haroz et al., 2016, 2017; Viduani et al., 2024). Thus, these culturally specific symptoms, such as (1) "crying," (2) "not speaking to anyone as usual" and (3) "staying/isolating at home," were added to the PHQ-8, resulting in an 11-item measure (PHQ-8 and the three items) used in the HIARA cohort study (Bolton, 2001; Bolton and Tang, 2002; Haroz et al., 2017). However, for this study, we only used the original eight items for the psychometric analysis. Therefore, future research assessing the dimensionality of the adapted PHQ-8 with these culturally specific symptoms should be conducted using an

A. One-factor (Depression) model (N=809)

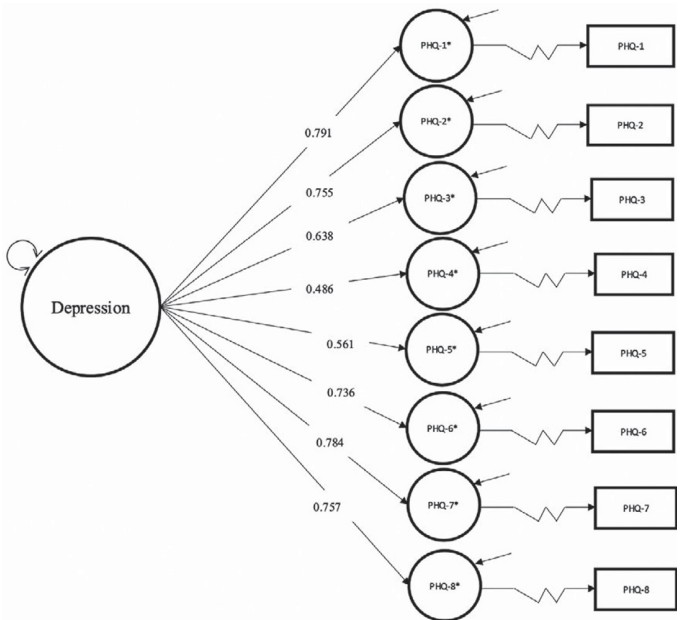

| Continuous Latent response variables | Observed ordinal variables | Items |
|---|---|---|
| PHQ-1* | PHQ-1 | Anhedonia |
| PHQ-2* | PHQ-2 | Depressed mood |
| PHQ-3* | PHQ-3 | Sleep disturbance |
| PHQ-4* | PHQ-4 | Fatigue |
| PHQ-5* | PHQ-5 | Appetite changes |
| PHQ-6* | PHQ-6 | Low self-esteem |
| PHQ-7* | PHQ-7 | Concentration difficulties |
| PHQ-8* | PHQ-8 | Psychomotor disturbance |

B. Two-factor (Somatic and Cognitive-Affective) model (N=809)

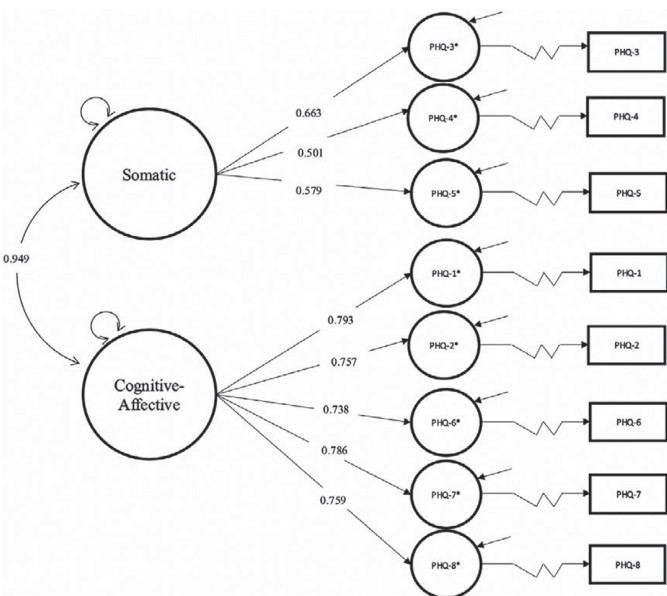

**Figure 4.** The one-factor (Depression) and two-factor (Cognitive–Affective and Somatic) models with the estimated standardized factor loadings using the DWLS estimator. The common factor variances were fixed to 1 (delta parameterization). The large curved bidirectional arrows represent the estimated correlation between the Somatic and Cognitive–Affective factors. The large circles represent the common factors. The small curved bidirectional arrows represent the variances of each common factor. The small circles represent the latent response variables. The unidirectional straight arrows represent the estimated standardized factor loadings. The short diagonal arrows indicate the residual variances of each latent response variable (small circles). The unidirectional "zigzag" arrows represent the set of estimated threshold parameters. The rectangular symbols represent the observed ordinal variables or indicators.

Exploratory Factor Analysis. That will help determine whether depression might have more than two factors (Lamela et al., 2020; Bianchi et al., 2022; Forbes et al., 2024), and a new theory of the dimensionality of depression could emerge from this.

Interestingly, symptoms related to weight gain and an increase in appetite were not mentioned by the participants during the qualitative studies. In contrast, the participants reported symptoms related to a decrease in appetite and weight loss (Table 1). It might indicate that gaining weight might be a sign of good health in that area, and an increase in appetite could be viewed as having a prosperous livelihood, which allows a household to eat more food. During our cognitive interviews with KIs, the word overeating of item 5, "poor appetite or overeating" of the PHQ-8, was frequently interpreted as being healthy. These two words are still lumped into a single item (5# *Over the last two weeks, how often have you been bothered by poor appetite or overeating?*) from the original PHQ-8.

**Table 5.** Measurement invariance across sex, ethnicity, education level and age group for the one-factor (Depression) PHQ-8 model

| Group | $\chi 2$ | df | p | RMSEA | 90% CI | CFI | TLI | ΔRMSEA | ΔCFI | SRMR | Decision |
|---|---|---|---|---|---|---|---|---|---|---|---|
| **Sex** | | | | | | | | | | | |
| Configural | 119.061 | 40 | <0.001 | 0.070 | [0.056–0.085] | 0.975 | 0.965 | | | 0.061 | |
| Thresholds | 128.151 | 47 | <0.001 | 0.065 | [0.052–0.079] | 0.974 | 0.969 | −0.005 | −0.002 | 0.061 | Accept |
| Thresholds and loadings | 129.448 | 54 | <0.001 | 0.059 | [0.046–0.072] | 0.976 | 0.975 | −0.006 | 0.002 | 0.062 | Accept |
| **Ethnicity** | | | | | | | | | | | |
| Configural | 137.074 | 80 | <0.001 | 0.060 | [0.042–0.076] | 0.982 | 0.974 | | | 0.069 | |
| Thresholds | 169.564 | 98 | <0.001 | 0.060 | [0.045–0.075] | 0.977 | 0.974 | 0.000 | −0.005 | 0.069 | Accept |
| Thresholds and loadings | 175.37 | 119 | 0.001 | 0.049 | [0.032–0.063] | 0.982 | 0.983 | −0.011 | 0.005 | 0.072 | Accept |
| **Education level** | | | | | | | | | | | |
| Configural | 165.282 | 60 | <0.001 | 0.085 | [0.070–0.100] | 0.963 | 0.948 | | | | |
| Thresholds | 184.418 | 74 | <0.001 | 0.078 | [0.064–0.092] | 0.961 | 0.956 | −0.007 | −0.002 | | Accept |
| Thresholds and loadings | 179.986 | 88 | <0.001 | 0.065 | [0.052–0.079] | 0.968 | 0.969 | −0.013 | 0.007 | | Accept |
| **Age group** | | | | | | | | | | | |
| Configural | 149.309 | 60 | <0.001 | 0.074 | [0.060–0.090] | 0.969 | 0.957 | | | 0.074 | |
| Thresholds | 167.266 | 74 | <0.001 | 0.068 | [0.055–0.082] | 0.968 | 0.963 | −0.001 | 0.006 | 0.074 | Accept |
| Thresholds and loadings | 193.404 | 88 | <0.001 | 0.067 | [0.054–0.080] | 0.964 | 0.965 | −0.004 | 0.002 | 0.083 | Accept |

*Note*: Group sample size. Sex (*N* = 809): Male *n* = 333, Female *n* = 476; Ethnicity (*N* = 809): Vezo *n* = 408, Masikoro *n* = 160, Antandroy *n* = 123, Other/mixed *n* = 118; Education level (*N* = 735): 0-year education *n* = 203, 1- to 5-year education *n* = 263, 6+ year education *n* = 269; Age (*N* = 809): Age group (16–29) *n* = 333, Age group (30–44) *n* = 251, Age group (45+) *n* = 225. CFI, Comparative Fit Index; RMSEA, root mean square error approximation; SRMR, standardized root mean square residual; TLI, Tucker–Lewis Index.

Had this item been separated, "poor appetite" would have been endorsed over "overeating." Although "overeating" might not be perceived as a depressive symptom in the BoR communities, individuals could still have an increase in appetite regardless of whether they have depression or not. Moreover, the context for overeating may differ from Global North cultural contexts, where overeating could be a symptom of fatigue or loss of interest in activities, and ready-to-eat products are easily accessible. In contrast, eating in Madagascar frequently involves cooking, which is quite effortful and may not align with typical depressive symptoms.

### *Probable prevalence of current depression*

We estimated that 8% of our study participants had probable current depression using the PHQ-8 score ≥ 10 in October 2023 (Kroenke et al., 2009). Our result was higher than the global estimated prevalence of depression (3.8%) from the WHO in 2019 and roughly twice the prevalence in Sub-Saharan countries, ranging from 3.4 to 4.9% (Gbadamosi et al., 2022). However, our finding was close to the probable prevalence of current depression in some European countries, as reported using the PHQ-8 (Arias-de la Torre et al., 2021, 2023).

### *Evidence of measurement invariance across demographic groups*

We found that all levels of MI were achieved across groups (i.e., sex, ethnicity, level of education and age category). This suggests that depression has the same interpretation for all groups. For instance, despite the cultural differences between the Vezo, Masikoro,

Antandroy and other/mixed ethnicities, depression has the same meaning among these ethnicities. This is supported by our qualitative findings, in which representatives from across these groups talked about local concepts of depressive symptoms in a similar way. This provides hope that scaling the PHQ-8 to other ethnic groups across Madagascar should be possible, especially if it is locally validated. Given that public health clinics across Madagascar are not yet using any mental health screening tools to detect depression, anxiety or other mental health disorders, it is possible that this study could help to provide a pathway toward a more generalizable assessment tool. The culturally adapted PHQ-8 can be integrated into the mHealth devices as a screening tool for depression used by nonmental health specialists, such as physicians, midwives, nurses and community health workers in the BoR. Nonmental health specialists can administer it to screen for depression, and they can refer a patient with high depressive symptoms to healthcare providers with the capacity to treat psychiatric disorders.

### *Limitations and future studies*

Longitudinal MI should be conducted for future studies to assess the stability of the psychometric properties of the adapted PHQ-8 over time. Although the HIARA cohort is a longitudinal study, we did not assess the stability of the psychometric properties of the PHQ-8 over time. Any longitudinal comparison of PHQ-8 mean scores within a group (e.g., females) may not be meaningful unless longitudinal MI is established (Liu et al., 2017).

A criterion validation study should be conducted to find an optimal cutoff to diagnose depression in the BoR using the PHQ-8,

supplemented by the three culturally specific symptoms, such as (1) "crying," (2) "not speaking to anyone as usual" and (3) "staying/isolating at home." These three culturally specific symptoms have already been collected in the HIARA cohort study. This is because using the total PHQ-8 score without the three culturally specific symptoms to screen for depression might provide biased estimates of the prevalence of probable current depression for the BoR communities (Fried and Nesse, 2015). Concern has also been raised about using the PHQ-8 standard cutoffs because they overestimated the probable prevalence of current depression (Levis et al., 2021). The adapted PHQ-8 should be mainly used to screen for depressive symptoms in the BoR. However, even without these validation efforts, this study provides a critically important tool for nonmental health specialists and researchers to screen for depression.

The culturally adapted and validated PHQ-8 should be limited to the BoR area. The main reason is that the local mental health syndromes in that area might differ from region to region in different parts of Madagascar. Thus, future research using our methodology should be recommended in other regions of Madagascar to enable the creation of a generalizable diagnostic measure.

## Conclusions

The adapted, translated PHQ-8 is a reliable and valid measure for screening depression in the BoR in southwestern Madagascar. Our study used a mixed-methods approach to culturally adapt and validate the PHQ-8 in Madagascar. We found local syndromes *Fiasan-doha* (general distress-like syndrome), *Alahelo maré* (depression/grief-like syndromes) and *Jangobo maré* (depression/grief-like syndromes) that included the main features of depressive symptoms.

**Open peer review.** To view the open peer review materials for this article, please visit http://doi.org/10.1017/gmh.2025.10032.

**Data availability statement.** Data requests should be addressed to the first author at hrandriamady@g.harvard.edu.

**Acknowledgments.** The authors would like to thank the Institut Halieutique et des Sciences Marines (IHSM), Professor Gildas Todinanahary, Emma Gibbons and Reef Doctor for their logistical support. In addition, the authors would like to thank Professor Dana McCoy at the Harvard Graduate School of Education for comments on an earlier draft of this manuscript. The authors would also like to thank Dr. Kathy Trang from the Department of Epidemiology, at the Harvard T.H. Chan School of Public Health for her assistance in interpreting the qualitative data, and Marie Celina Razanajaosoa for her help in backtranslating the measure. We are grateful to Dr. Aina Le Don Nomenisoa for helping us produce the HIARA study map. The authors are grateful to Dr. Fabien Rakotondramanana from the Ministry of Public Health in Toliara. The authors would like to thank Dr. Vola Nirina Andrianavalona and Dr. Nivohanitra Razafindrasoa from the Ministry of Public Health in Antananarivo for conducting the initial focus group discussions. Above all, the authors would like to thank all participants in the study and the Bay of Ranobe communities.

**Author contribution.** H.J.R.: Research conception and design, collection of data, analysis of data, interpretation of data, writing the manuscript, review and editing the manuscript. M.S.: Interpretation of data, writing the manuscript, review and editing the manuscript. R.E.S.: Research conception and design, interpretation of data, review and editing the manuscript. A.F.M.: Research conception and design, collection of data, analysis of data, review and editing the manuscript. R: Collection of data, analysis of data, review and editing the manuscript. M.R.: Collection of data, analysis of data, review and editing the manuscript. N.V.V.: Collection of data, analysis of data, review and editing the manuscript. F.D.: Collection of data, review and editing the manuscript. M.Y.S.: Collection of data, review and editing the manuscript. J.C.M.: Collection of data, review and editing the manuscript. H.O.R.: Collection of data, review and editing the manuscript. K.C.K.: Research conception and design, interpretation of data, writing the manuscript, review and editing the manuscript. C.D.G.: Research conception and design, collection of data, interpretation of data, writing the manuscript, review and editing the manuscript.

**Financial support.** Financial support for this study was provided by Belmont Forum through the National Science Foundation (RISE-2022717 CDG) and the Harvard President's Climate Change Solutions Fund (CDG and KCK).

**Competing interests.** The authors declare none.

**Ethical statement.** All participants were recruited and enrolled following our IRB-approved study (Protocol #20–1944 and 22–0491, Committee on the Use of Human Subjects, Office of Human Research Administration at the Harvard T.H. Chan School of Public Health). The study was also reviewed and approved by the Ethics Committee of the Ministry of Public Health (N036MSANP/SG/AMM/CERBM) and subsequently reviewed and stamped by the Division of Mental Health Services at the Malagasy Ministry of Health, as well as by the local medical inspector in Toliara II.

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
