## [Reviewer Report]

In this paper the authors evaluate whether an adapted PHQ-8 would be suitable for use to measure Depression in southwestern Madagascar. Given the paucity of work evaluating mental health tools in Madagascar, this is a timely and important piece of work. The authors combined qualitative work with quantitative measurements and worked with local communities to develop terms and idioms appropriate to capture the symptoms described in the PHQ-8. This is a thorough and very well performed project and I only have minor comments.

Methods:

1) Page 6: The focus groups were comprised of adolescents and adults, do you have their ages? It would be interesting to know to what extent there was agreement between the participants relative to their ages?

2) Page 7: I believe the free listing interviews were conducted in a group (rather than individually)?

1) Quantitative study: who delivered the PHQ-8 to the study participants? Were these field workers or members of the team who also conducted the interviews? How / where was the testing done? If members from the same household were tested, did they do this separately? Did the participants who took part in the qualitative study section also take part in the quantitative study?

Results & Discussion:

2) While it is maybe not surprising that some factors were not mentioned (e.g. weight gain), it is curious that ‘increase in appetite’ was not mentioned (Table 1), given the reported level of food poverty in Madagascar. This is touched upon in the discussion, but I wonder if this means that appetite related questions may not be very informative when measuring mental health (if appetite in general is associated with good physical health or being prosperous).

3) P.24: if participants who took part in the qualitative study also took part in the quantitative study, were those who mentioned psychotic features (e.g. self-talk, talk-nonsense…) also people who had a PHQ-8 score above 10?

You report measurement invariance across the groups (p.22), it would be useful to know what the prevalence of participants with PHQ-8 scores >10 is within the different age / sex / community groups. I realise this isn’t the point of the study, but it would be useful to know how prevalent depression is within the different groups, notably as you touch on this in the probable prevalence for current depression in the Bay of Ranobe.

---

## [Reviewer Report]

Thanks for the opportunity to review this well-written, useful, and interesting manuscript. There is a clear need for validated measures for use in Madagascar, and particularly in the South, given the lack of available, locally contextualised or developed measures for use in this region. Although overall I’m very positive about this manuscript, I have a few suggested changes.

First, in the impact statement, there’s a discussion about the use of the PHQ as a tool to refer people for treatment in a psychiatric clinic, but this seems like an unlikely use case for the measure given the extreme lack of availability of psychiatrists in the country and particularly in the Southwest.

Second, although the PHQ-8 hasn’t been validated except in two countries in Africa (and now Madagascar), the PHQ-9 is well-used across Africa. Tthere should be a discussion of this in the Introduction, and on its validity and reliability in African contexts, as well as on any use in Madagascar for the PHQ-8 or PHQ-9 previously.

Third, it would be useful to provide an explanation of why the focus of this paper is on the Bay of Ranobe in particular.

Fourth, more details are needed about the conduct and participants for the focus groups and interviews. The manuscript should include the mean length and the standard deviation of the focus group discussions, the free listing interviews, and the key informant interviews.The exact number of participants in the focus groups should be specified. The age of the participants in the focus groups should also be included. I see it says adolescents and adults, but it’s not clear what is meant by adolescents in this study nor why adolescents and adults were included in these focus groups together. Further, there should also be more information on the language used, and on the translation and transcription process for the focus groups and for the interviews. It would be useful to specifically mention here any uniquenesses about the dialect in the Bay of Ranobe area. More information on the process of the thematic analysis should also be included.

Fifth, for the survey study, it would be useful to have information on how the random sampling was conducted as well as how the sample goes from 1539 total to 809 survey study. Is that because the other 730 participants are adolescents and therefore not included in this manuscript?

Given that the focus group and interviews suggested there were three different syndromes which are associated with depression in this region (Fiasan-doha, Alahelo maré, Jangobo maré), why was the decision made to do a two-and-one factor confirmatory factor analysis? Why not look for three factors?

I very much appreciate the comprehensiveness of having conducted the focus groups and the two types of interviews in order to understand local conceptions of these syndromes and of depression. However, given that ultimately there is very little difference between the PHQ-8 and the translated/locally contextualised version of the PHQ-8, as evidenced by the translation/back-translation process having “no major differences” between translated and back-translated version, it would be useful to critically analyse whether this type of process is really necessary. An extended discussion of this in the Discussion section would be valuable. Relatedly, there is a bit in the Discussion about how crying and components of that are unique aspects of depression in this context, but then they haven’t been added to the 8-item scale and instead you’ve just added them to your larger cohort study. Why is that? Why not include them in the locally developed or adapted PHQ-8? Similarly, given that you haven’t included this concept of depression within the PHQ-8, do you think that the cut-off score and the 8% of the population with probable depression is accurate? If you’re not including all of the local aspects of depression, then perhaps this is not going to be an accurate representation of the proportion of the population with depression, nor will the cut-offs be appropriate for this population.

Finally, this is a very small point, but it would be useful to provide some clarity on what the hunger index of 36.3 means in practise. It would also be good to provide the hunger index for the Southwest specifically.

---

## [Editor Report]

May you kindly address the reviewer comments, particularly providing a more nuanced rationale for the selection of the PHQ-8. Additionally, ensure synergy between the impact statement, methodology, analysis, and conclusions, while also clarifying the methodological queries raised.

---

## [Reviewer Report]

The authors have addressed my concerns. I am satisfied with the added changes and believe that this is an important piece of work that will be useful for many researchers.

---

## [Reviewer Report]

I have two minor changes to suggest:

1. When discussing the first focus groups, the age range is people 16 to 22 years old. There’s a discussion of the value of having people who are adolescents and adults from different age ranges, but 16 to 22 is a very limited age range. Indeed, many researchers including some who work in Madagascar would view everyone in this age range to be an adolescent (e.g., Hadfield et al., 2025; Sawyer et al., 2018).

2. Although the manuscript now includes a description of the Bay of Ranobe, it does not give a clear rationale for why this area was chosen to examine the PHQ-8 / why you were specifically interested in mental health in this area.